# Security and Privacy in Connected Vehicle Cyber Physical System Using Zero Knowledge Succinct Non Interactive Argument of Knowledge over Blockchain

Rabia Khan [1], Amjad Mehmood [1,2,*], Zeeshan Iqbal [1], Carsten Maple [2] and Gregory Epiphaniou [2]

1 Institute of Computing, Kohat University of Science & Technology, Kohat 26000, Pakistan
2 Secure Cyber Systems Research Group (CSCRG), WMG, University of Warwick, Coventry CV4 7AL, UK
* Correspondence: dramjad.mehmood@ieee.org

**Abstract:** Security has been the most widely researched topic, particularly within IoT, and has been considered as the major hurdle in the adoption of different applications of IoT. When it comes to IoV, security is considered as the most inevitable component to ensure a safe and smooth driving experience. CAV is the new era of transportation, integrating intelligence and self-driving capabilities within vehicles and that requires strong security measures to ensure safety. Security alone is not enough. Instead, a complete package including privacy of the vehicles and passengers needs to be added in addition to secure communication. This is because CAVs are under continuous cyber threats and attacks and the most important among them is the DDoS, where a remote attacker can hijack/launch attacks on vehicles remotely. Single point of failure attacks target the centralized trusted body in order to mislead the connected vehicles for personal gains. In this paper, the authors have proposed a secure communication system for CAVs using blockchain, which also ensures the privacy of the vehicle/people. The paper highlights the major components of the proposed system, and its performance is evaluated to check its efficiency against DDoS and Eclipse attacks. The unlinkability and anonymity of the vehicles have been ensured using the zk-SNAKR protocol over Blockchain.

**Keywords:** blockchain; zk-SNARK; connected and autonomous vehicles; cyber physical system; unlinkability; anonymity

## 1. Introduction

Advancement in communication technologies and Artificial Intelligence has had a profound impact on the lives of people. Connected and Autonomous Vehicles (CAVs) are one such example. Connectivity and automated technology can shape the driving experience when CAVs will be capable of replacing humans. Other sister technologies such as advanced sensor networks, GPS, remote processing, and telecommunications networks can all be used to achieve the purpose. CAVs have the potential to reduce traffic accidents, improve modes of transportation and enhance quality of life [1]. They can bring a revolution to the locomotive industry. Researchers in the domain have highlighted many benefits and positive aspects of CAVs within the locomotive industry in general and in human life, including shortened distances not raising delay times, reduction in vehicle premiums, the smaller size of traffic departments, and reduction in emergency patients due to reduced traffic accidents [2]. CAVs are one of the applications of CPS within IoT, which have transformed the driving experience. They can provide a high level of safety and anticipate the traffic conditions on roads because of a high degree of autonomy, which reduces the burden on drivers [3]. CAVs use sensors and wireless sensor networks to obtain relevant traffic-related and other important data/information. The rising demand for CAVs brings along opportunities and challenges. The major challenges being observed in the realization of CAVs are security and privacy. Travelers' private data and communication

between CAVs are valuable assets both for passengers and transportation companies to ensure secure traveling. People expect to keep their private data, such as vehicle registration numbers and the registration data of people who are known to the national certification authority and are responsible for the registration of vehicles and people, confidential. People are concerned about the privacy protection of their data during communication among different CAVs and infrastructure. Personal privacy is very easy to be hacked as an attacker can easily obtain access to users' information, such as text messages, phone numbers, call logs, etc. and even driving habits can be invaded [4]. Therefore, the first challenge to be faced by CAVs is that no one except the certification authority should have access to the private data of people and vehicles, to avoid tracking. The second challenge is to ensure that a valid vehicle can communicate with other CAVs with proof that it meets certain requirements without revealing its privacy. This will guarantee the security of the vehicles and the data being communicated. It is quite easy for the cyber attacker to temper and forge the communicated data, e.g., if the information related to the road condition is purposely forged by the attacker, it may be life-threatening to the security of the travelers [4].

Blockchain is considered a promising solution to the challenges and problems mentioned above as it has been applied in different scenarios, such as virtual currency, e-government, drug supervision, intelligent healthcare, etc. [5]. Blockchain is a distributed ledger of immutable transactions that builds trust among numerous least- trusted nodes without the involvement of a third party and ensures that all the nodes within the blockchain network verify and share data using a certain consensus algorithm [6]. Working with blockchain is quite simple. It works similarly to a peer-to-peer network where a user starts a transaction, and once done, a block is assigned to the transaction. This block is then broadcasted to the blockchain network, where all the nodes acquire the information. The block gets mined and validated and is added to the chain [7]. Smart contracts allow for the least trusted parties to communicate with each other by automatic verification of the scripts within the blockchain [8]. The first challenge will be addressed using blockchain in this research. Blockchain commonly comes under network attacks such as DDoS and Eclipse, so this research will assess the performance of the proposed system against DDoS and Eclipse attacks.

Zero-Knowledge Succinct Non-Interactive Argument of Knowledge (zk-SNARK) is a protocol that creates a framework where a party known as "Prover" can convince another party known as a "Verifier" that he/she knows certain information without revealing that information. zk-SNARK uses the concept of zero-knowledge proof. zk-SNARK can be explained with a simple example. To develop a proof of a transaction between two parties, at least one of the two parties must have complete information about the transaction. In the traditional proof system, the proof (password) is compared with the password the user enters to access some online system. In any online system, the user enters a password and the network checks the password for its verification. The network must have access to the user's password, and only then password verification can be completed, and the system can verify if the user has entered the correct password or not. On the other hand, in the zk-SNARK proof system, the same example can be demonstrated in a manner where the user would ensure the system that he/she owns a correct password without revealing the password. Therefore, zk-SNARK ensures security and privacy in case the system does not save the password, so it cannot be stolen by any unauthorized access to the system. zk-SNARK can be applied to blockchain transactions as it hides the sender and receiver of the transaction resulting in privacy-preserving, which is going to be achieved in this research. In this paper, the security of the user's data and user/vehicle first place is proposed using blockchain technology while the privacy of user/vehicle identification is preserved using zk-SNARK. The conceptual diagram of the proposed architecture is represented in Figure 1.

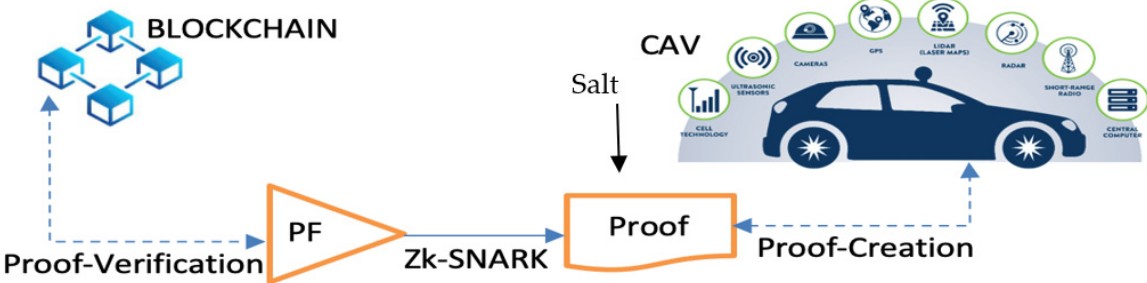

**Figure 1.** Conceptual Diagram of Proposed System.

The goal of this paper is to provide a blockchain-based privacy-preserving and secure data communication mechanism among CAVs using the zk-SNAKR protocol. This scheme also intends to ensure the confidentiality, integrity, and availability of data. The main contributions of the paper can be summarized as follows:

1.  A secure decentralized data sharing scheme has been proposed based on blockchain to achieve privacy-preserving;
2.  The proposed scheme utilizes zk-SNARK, in which users can prove that they meet certain requirements imposed for valid communication without revealing their identities. The requirements are set by the centralized authority and are saved in a smart contract where the user constructs a zero-knowledge proof ($\pi$) and submits it to RSU for verification. Once verification has passed, the user will be capable of starting communication with other vehicles;
3.  The analysis of the proposed scheme and performance evaluation on specified attack vectors (DDoS and Eclipse) demonstrates that the proposed scheme achieves the security and privacy of the users and data.

The rest of the paper is organized as follows: Section 2 is the literature study, Section 3 is problem formulation, Section 4 discusses the Proposed System, Section 5 is the performance evaluation of the proposed system, Section 6 is the discussion part, and Section 7 is the conclusion.

## 2. Literature Study

*Different security and privacy solutions have been* proposed for the CAVs both with and without using blockchain technology. This literature review will highlight the major contributions of both. CAVs are susceptible to both active and passive cyber attacks. During active attacks, the main intention of the attacker is to change/destroy or modify the system. Direct attacks bring more damage to CAVs security and pose a high threat to privacy protection, whereas passive attacks read data communicated between a vehicle and destination such as RSU, which poses a lower threat as compared to active attacks. Passive attacks are generally quite difficult to identify because attackers do not change the contents of the message, since both sender and receiver are unaware of the man in the middle [9]. Types of direct attacks are message spoofing, replay attacks, DDoS, and Eclipse attacks, whereas passive attacks are eavesdropping/release of information and traffic analysis.

Cryptography is the most commonly used security and privacy-preserving technique used for CAVs. The literature shows different studies where cryptography has been used to propose security and privacy solutions. A secure and authenticated key management protocol (SA-KMP) using symmetric keys is used to reduce the computational cost as compared to asymmetric cryptography [10]. The trust cooperative transmission protocol for multiple-hop broadcast has been proposed, which selects the best relay that minimizes the function of a finite number of metrics among all the relays [11]. An anomaly-based intrusion detection system (IDS) known as Clock-based IDS has been implemented for measuring and exploiting the intervals of the systematic and periodic in-vehicle messages to fingerprint the ECUs [12]. A light-weight intrusion detection algorithm for in-vehicle networks works effectively and is based on the analysis of the time interval of Controller

Area Network (CAN) messages [13]. An identity verification technique has been introduced based on the blockchain using symmetric and asymmetric encryption [14]. The proposed system requires an extensive verification of vehicles, RSUs, and blocks before being part of blockchain. The proposed technique is effective in providing privacy protection but at the same time is quite time-consuming. A blockchain-based decentralized platform is introduced with the name EVChain for sharing charging credits within the electric vehicle (EV) charging market [15]. The main intention of the proposed system is to share the charging credit by hiding the real identities of vehicles' owners through the k-anonymity privacy-preserving technique. However, since the model utilizes different blockchains, it may require some time for processing the user requests.

The intrusion Detection System (IDS) for VANETs works to detect false information attacks using statistical techniques quite accurately [16], but it does not take into consideration the communication data. Tri-Blockchain architecture has been proposed for intelligent vehicular communication [17]. The proposed scheme is mainly designed for communicating highway accident cases to nearby vehicles and the authorities. It is based on a consensus mechanism, which means that information is verified by many vehicles and becomes part of a block to be added to the blockchain. However, a group of malicious vehicles may be the part of the consensus process, allowing for false traffic information to be communicated to nearby vehicles and the blockchain. This can make the proposed scheme less reliable. Blockchain-based secure device-to-device communication architecture is proposed using the Bloom Filter [18]. The proposed system is intended to prevent cyber attacks in military networks. Every vehicle is required to be registered with respective unmanned aerial vehicles (UAV) using their respective MAC addresses and in return, an alias is generated for them. The alias is used for communication instead of real identities. Bloom filter is probabilistic in nature, which means at times it can generate highly false-positive results. This issue is tackled by limiting the number of hash functions. Still, the processing time of hash functions is quite long in the proposed architecture, making it less effective, particularly for military networks. A privacy-preserving data sharing technique involving federated learning has been proposed based on blockchain [19]. The proposed system depends on machine learning to train a global data model. It is used for two types of transactions, i.e., retrieving data and sharing data. Because of storage limitations and computation resource restrictions, the original data is stored with the owners of the data while the identity of data providers is stored on the blockchain. The proposed system works effectively in data sharing but it does not reveal how the identity of data providers can be preserved.

Limited efforts have been observed in the literature that propose security and privacy solutions based on blockchain. Automotive security and privacy framework using blockchain based on changeable private keys is proposed [20], but it relies heavily on the traditional encryption algorithm. An open platform for exchanging messages between service providers and drivers based on blockchain [21] has been proposed. A self-pseudonym generation technique is introduced. The pseudonyms are stored on the blockchain to ensure the security and privacy of the vehicular network [22], but the proposed system relies on the traditional Ring Algorithm, consuming a lot of time in computation. The privacy-preserving mechanism for multimedia sharing has been proposed based on blockchain that depends on pseudonyms for hiding the identities of vehicles, roadside units, and users [4], but it relies on the centralized trusted authority for the issuance of pseudonyms. A decentralized blockchain-based architecture for VANET has been proposed for tackling distrust among vehicles and infrastructure [23]. The proposed system uses a dynamic encryption technique known as threshold encryption and k-anonymity unit algorithms. The k-anonymity unit algorithm takes time to calculate sub-identities for vehicles, making the proposed system less effective in real-time scenarios. A blockchain-based authentication protocol has been proposed for the cooperative VANET utilizing the digital signature algorithm RSA-1024 [24]. The proposed system is claimed to ensure the confidentiality, integrity, non-repudiation, and privacy protection of IoVs. It successfully achieves privacy

protection and the security of IoVs. However, the performance of the system becomes affected because of the use of a cooperative transmission protocol. The throughput of the system is reduced further if the number of vehicles continues to increase. Signatureless public key infrastructure has been implemented on the blockchain, relying on message ratings and credibility to mitigate network attacks [25]. The performance of the proposed system is quite a stack since the authentication process takes a lot of time. It requires authentication from different blockchains used within the proposed system. This reduces the communication setup and, hence, produces delays in valid communication.

A privacy protection system has been proposed based on blockchain for IoV that has two-way authentication with a key agreement algorithm designed on random numbers [26]. It ensures the confidentiality of the security information. However, the proposed system relies on the PoW consensus algorithm, which consumes a lot of computing resources, forcing the system to create undesired delays in generating responses. A decentralized and location-aware traffic management system has been proposed for protecting data integrity and privacy using Zero-Knowledge Range Proof (ZKRP) based on multiple blockchain environments [27]. The proposed system introduces the concept of gateways, which reside in between two adjacent blockchains that help in switching the traveling vehicles from one blockchain to another. This means the gateways are responsible for the authentication of vehicles and ZKRP is implemented within gateways. Although the proposed system claims to achieve security and privacy, it does not illustrate the mechanism for vehicles' registration and authentication and how ZKRP is generated. The privacy-preserving fair exchange scheme (Vehicle to Grid Exchange) V2GEx has been proposed for Vehicle to Grid (V2G) and is based on a blockchain, utilizing zero-knowledge funds for fair funds deposit and claims [28]. The proposed system makes use of multiple zk-SNARKS for fund deposits and claims, causing the system slow processing since proof generation is an extensive task. A distributed firmware update scheme is proposed for autonomous vehicles using blockchain and zero-knowledge proof to generate proof of distribution [29]. Smart contracts are used for firmware updates. The updates are added to the smart contract as a transaction by the manufacturers, defining the access policy for vehicles in such a way that a vehicle meeting the criteria, defined in the access policy, can obtain updates. For vehicle authentication, attribute-based encryption (ABE) is used in the proposed system. The proposed system relies on ABE for the justification of access policy but since ABE is quite time-consuming, it can make the overall transmission process slow albeit secure. A blockchain-based security and privacy system named "BPAS" has been proposed for VANET to ensure the trustworthiness and accuracy of messages within VANET in addition to protecting their privacy [30]. The proposed system provides single registration for vehicles through Trusted Authority (TA) and the registration information is saved in the onboard unit (OBU) of the vehicles. If a vehicle wants to be part of the proposed network, it has to be authenticated from its OBU. The proposed system also relies on attribute-based encryptions (ABE) and a biometric extraction Fuzzy extractor. It works on the assumption that OBU is temper-proof, which can be risky.

A blockchain framework is proposed for the security and transparency of the connected vehicles and users by recording every activity of all the entities within the blockchain network [31]. It is to ensure secrecy and transparency between customers and cab drivers. The registration and authentication of the vehicles is done using IoT device numbers within the vehicles and their identifiers. The proposed system is quite obscure and relies on the IoT devices' identifiers installed within the vehicles. A multi-agent AIM (MA-AIM) system is proposed based on Vehicle-to-Vehicle/Vehicle-to-Infrastructure (V2V/V2I) communication using blockchain for the security of vehicles crossing through an intersection [32]. The proposed system uses the concepts of gateways (Intersection Management-IM) to authenticate the vehicles' information by allowing them to cross the intersections. The proposed system is quite open in its working which means any vehicle can send messages/requests to intersection management (IM) by using its registration number. Hence any malicious

vehicle can easily mislead the IM, resulting in traffic congestion near the IM. A summary of the literature study is presented in Table A1 in Appendix A.

The work more closely related to ours is [33,34]. Both are different from one another in terms of the system being introduced and the design goals. The system proposed in [33] is mainly for the fair dissemination of ads within a vehicular network using blockchain. It uses ZKPoK, a variant of ZKP to hide the identity of vehicles, but it does not take into consideration the RSUs. The proposed system incurs heavy computational costs during registration, ad dissemination, and reward payment. The major computational cost relates to ZKP, which is quite a heavy variant of ZKP. The overhead is also observed for on-chain computation as compared to off-chain computation. The proposed system is by large based on honest assumptions such as the honest working of RSUs to prove the correctness of the system. As opposed to [33], our proposed system uses zk-SNARK, which is an efficient and light variant of ZKP as compared to ZKPoK. No honest assumptions were taken into consideration while designing our system. It is required that every vehicle as well as RSU should be registered within the network before participating in the communication process. The off-chain computation in our proposed system requires the storage of vehicles/RSUs identity on a Merkle tree, resulting in minimal computational costs, whereas on-chain computation involves proof generation, which is also less since the proof is generated once. RSU is not assumed to be equipped with heavy powerful resources in our proposed system for economic purposes.

The proposed system in [34] is a design for on-street parking authentication using zk-SNARK. Though the construction of proof in our proposed system is similar to [34], there are basic differences in the design goals of both systems. The proposed system in [34] is dependent on a centralized server, which can result in a single point of failure while our proposed system is a decentralized architecture, removing the option of a single point of failure. The proposed system in [34] has limited connectivity, utilizing only Bluetooth, which limits the communication range, whereas our proposed system uses multiple communication technologies including Bluetooth, which enhances the communication range depending on the range of RSUs.

## 3. Problem Formulation

The system model of the proposed system is shown in Figure 2.

The proposed system combines blockchain, smart contract, and zk-SNAKR to achieve secure data communication and the privacy-preserving of individuals and vehicles. Four major elements are involved in this system model: (i) Vehicles, (ii) CA, (iii) RSU, and (iv) blockchain. Their functions are described as follows:

4. Vehicles: CAV acts as a prover in this system who wants to share data with RSU or any other CAV. The prover requests the registration identity from the certification authority to take part in valid data sharing/communication.
5. CA: is mainly a trusted Government organization that is responsible for issuing registration identities to vehicles and users. CA is also responsible for the specification of the requirements/function within the smart contract that needs to be fulfilled by the CAV before allowing it to start data sharing with another CAV.
6. RSU: major roadside infrastructure that is responsible for authorizing the identity of CAV and signaling all other CAVs, allowing for the authenticity of a particular CAV before allowing it to start data sharing.
7. Blockchain: stores smart contracts which are rules/requirements/functions to be fulfilled by any communicating CAV. It is also responsible for storing the communication between CAVs as transactions and is distributed over the blockchain network.

A vehicle (prover) requests CA for registration and sends the identity information such as the registration number of the vehicle to CA. CA is responsible for the initialization of the system and the proof verification that allows a CAV to enter the network and start communicating with other CAVs. Once the registration identity has been received by the prover, it sends the request along with a proof 'π' to CA, for verification of its identity

and asks for permission to allow communication. The smart contract approves/rejects the requests for the vehicle by matching them with the information stored in the Merkle tree by the CA and responds to CA and also nearby RSU. RSU then signals all neighboring vehicles and RSUs about the status of the prover and also allows the prover to start communicating with other CAVs if its status is verified. The prover starts communicating with neighboring CAVs and the communication is monitored and saved by the nearby RSU. That communication is then stored on blockchain as a transaction which is then distributed over the Blockchain network for all the nodes to save/update the record. List of acronyms is provided in Table 1.

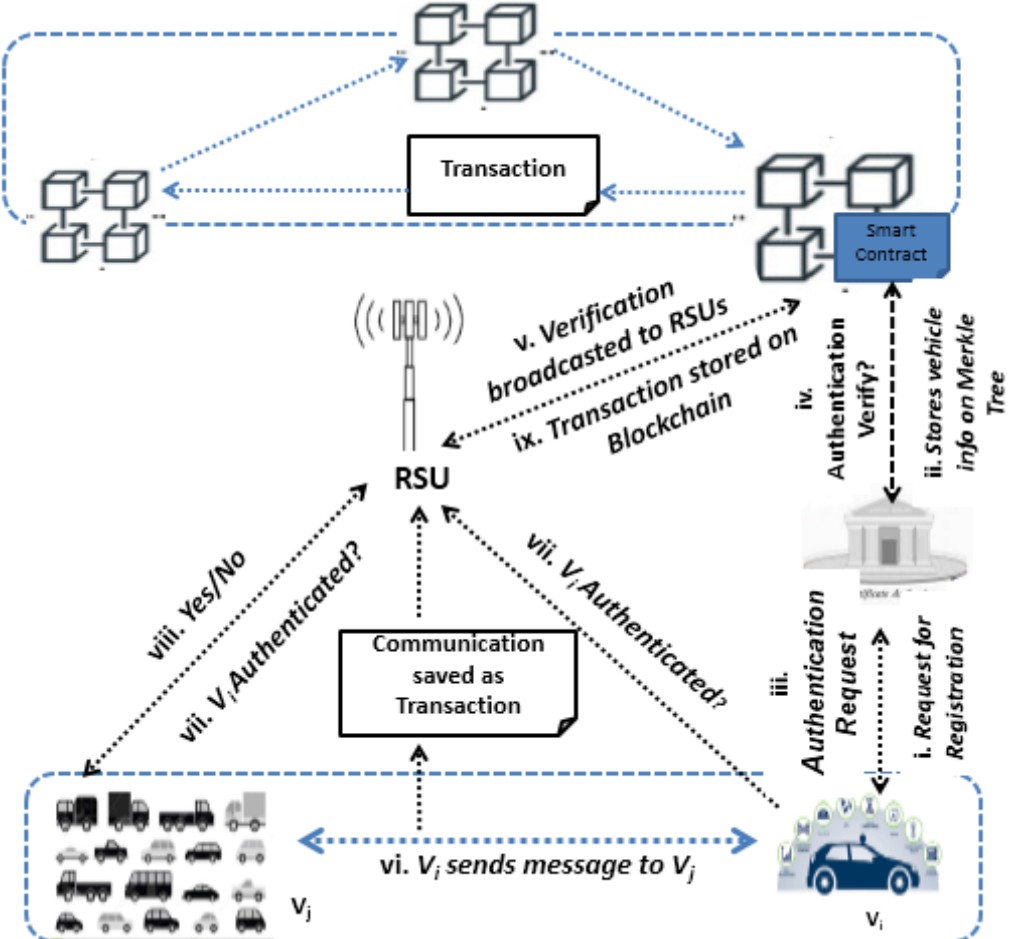

**Figure 2.** Security and privacy using zk-SNARK over blockchain-proposed architecture.

The following design goals are to be realized under our proposed model:

Goal (1) Anonymity and Unlinkability: The privacy of vehicles needs to be preserved, which means the vehicles must be anonymous such that any malicious RSU/vehicle or the cyber attacker may not trace the real identity of vehicles. In addition to this, the driving trajectory, driving history, and driving aptitude must not be traced by any malicious entity such that the location of a particular vehicle may not be inferred.

Goal (2) Security against Single Point of Failure: The proposed system must resist a single point of failure attacks.

Goal (3) Security against DDoS and Eclipse Attacks: The proposed system must resist DDoS and Eclipse attacks.

Goal (4) Tracing Malicious Entity: Vehicles are more concerned about their privacy and not just security when they require entering some network. Anonymity is required to protect the real identities of CAVs so that specific vehicle may not get hit. However, a malicious CAV needs to be detected, which creates a challenge for the proposed system.

**Table 1.** List of Acronyms.

| Acronym | Definition |
| --- | --- |
| CAV(s) | Connected and Autonomous Vehicle(s) |
| CPS | Cyber Physical System |
| IoT | Internet of Things |
| zk-SNARK | Zero Knowledge Succinct Non-Interactive Argument of Knowledge |
| RSU(s) | Roadside Unit(s) |
| DoS | Denial of Service |
| ECU(s) | Electronic Control Unit(s) |
| VANET | Vehicular Adhoc Network |
| IoV | Internet of Vehicles |
| CA | Certification Authority |
| $\pi$ | zk-SNARK proof |
| w | Witness |
| x | Statement/argument |
| DDoS | Distributed Denial of Service |
| $Sig_i/sig_j$ | Signature of vehicle i/j |
| $PK_i/PK_j$ | Public Key of vehicle i/j |

## 4. Proposed Solution

The fundamental primitives used in the proposed system are introduced here.

### 4.1. Blockchain

The increasing security concerns within IoV make it difficult to realize it as a reality. Centralized security measures are being adopted but they are not as effective as they should be, so the research community is now moving towards decentralized security approaches. Blockchain technology is the most suitable and ideal approach to providing a decentralized means of communication within IoT, particularly for IoV [35]. Blockchain is a distributed ledger that is maintained by distrusted miner network nodes. These nodes mutually reach an agreement through some consensus protocol such as proof-of-stake and proof-of-work. Blockchain has major characteristics which distinguish it from other technologies and these are: (1) *Correctness and Traceability.* Blockchain is a transparent ledger that encourages every node to trace and verify the correctness of data that arrives for storage on it. (2) *Irreversibility and Immutability.* Once the data is recorded on blocks, it is hard to format or temper it since every block is saved using hash codes in such a manner that the hash of the previous block is stored in a current block along with its hash, which ensures irreversibility and immutability [33].

There are two types of blockchain: online/public [36,37] and offline/private blockchains [36,38]. The results of transactions are stored on an online blockchain e.g., the output of a smart contract [36], resulting in the public part of the blockchain while offline blockchain takes the transaction value outside of the blockchain, forming the private part of blockchain. In our proposed system, the Merkle tree is stored in a public blockchain whose updates are constantly being distributed among all the nodes within the blockchain while the vehicle identity is stored in a private blockchain. It is also important to mention here that every communication between CAVs is not required to be stored as a transaction since it increases the computation time in validation and authentication. Instead, the malicious communication/behavior of the CAV/user is required to be stored as a transaction. The structure of the blockchain is represented in Figure 3. Blockchain has been proposed as a mechnaism for annonimity and decentralization of information within VANETs [39]. Blockchain has been integrated smartly with the Internet of Vehicles (IoV) from the perspective of the Intelligent Transportation System (ITS) [40]. This is because of the decentralized management and strong secuirty measures. Blockchain has been made to be light-weight with the complete features of the normal blockchain to reduce the reliance on computing

resources [41]. The light-weight blockchain has been proposed for IoV, which reduces the delay in a blockchain query within the IoV system.

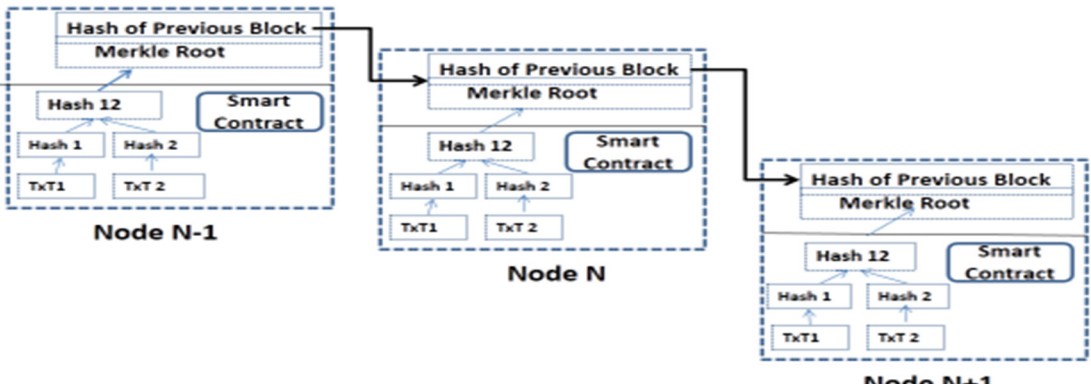

**Figure 3.** Structure of blockchain.

### 4.2. Smart Contract

A smart contract is a program that is automatically executed within a secure environment if specified conditions are satisfied. Smart contracts have been implemented on Blockchain for designing decentralized applications, particularly when dealing with secure communication within the IoT. In our proposed system, the smart contract will be used for the verification of zero-knowledge proofs to decide if a CAV should be allowed to communicate within the proposed system or if it should be blocked.

### 4.3. Zero-Knowledge Proof System (ZKPS)

Zero-Knowledge Proof System enables a user to prove the validity of a statement without revealing anything about the statement [42,43]. The variant of zero-knowledge proof used in this proposed system is zk-SNARK. In zk-SNARK, a prover can prove the validity of his/her statement (identity) without revealing details about personal information. For example, the prover wants to prove that he has a valid registered identity number for both himself and his vehicle without revealing the identity number. The verifier who is RSU in this case should learn that the statement of the prover is true without knowing the identity of the prover. zk-SNARK is a non-interactive variant of a zero-knowledge proof system, providing constant-sized proof and also results in the fast verification of statements [36]. zk-SNARK has been actively researched in recent years [42–44]. zk-SNARK is an efficient protocol that gives a user complete control over the way of how he/she proves his/her identity [28–36]. zk-SNARK plays an important role in proving the correctness of user statements in the proposed system. zk-SNARK has been proposed to be in Internet of Things (IoT) and its applications. zk-SNARK in collaboration with blockchain as a distributed ledger provides privacy protection and security [45]. zk-SNARK is an efficient protocol that gives a user complete control on the way how he/she proves his/her identity [28–35]. zk-SNARK plays an important role in proving the correctness of user statement in the proposed system. zk-SNARK has the following characteristics:

- Completeness: If the prover acts honestly and sends a true input statement, then the honest verifier following the zk-SNARK protocol correctly will be convinced of the statement.

Our proposed system consisting of functions (Setup, Prove, Verify) is complete for R if for all R $\epsilon$ R($1^\lambda$) and for all (x,w) $\epsilon$ R:

$$P[\![(srs) \leftarrow Setup(R,C); \pi \leftarrow prove(srs,x,w) : Verify(srs,x,\pi) = 1]\!] = 1$$

Witness 'w' is kept secret and in normal circumstances is not known to both the prover and the verifier. It helps in the generation of proof during the zk-SNARK proof setup.

- Soundness: If the prover tries to cheat by sending a false statement, the verifier will not be convinced that the statement is true, rejecting the proof with high probability. The proposed system is sound as far as valid proof is provided, so it is possible to extract a valid witness.

$$\text{Adv}(\lambda) = \text{P}[\![(\text{srs}) \leftarrow \text{KeyGen}(\text{R},\text{C}); (x,\pi) \leftarrow \forall(\text{srs})|\text{Verify}(\text{srs},x,\pi) = 1, \ (x,w) \notin \text{R}]\!] \leq \text{neg}(\text{n}))$$

The argument System $\text{Adv}(\lambda)$ is defined as follows and is completely sound for any adversary within the proposed system, and if there exists an extractor 'x∀' such that $\text{Adv}(\lambda) \approx 0$:

$$\text{R} \leftarrow \text{R}(1^\lambda), \ \text{Setup} \ (\text{R},\text{C}) \rightarrow \text{srs}; \ \forall(\text{srs}) \ \rightarrow (x, \ \pi); \ w \leftarrow x\forall(\text{Trans}\forall); \ \text{assert}(x,w) \notin \text{R}$$

- Zero-Knowledge(ness): If the statement sent by the prover is true, the verifier learns nothing beyond the fact that the statement is true. Therefore, the proof is zero-knowledge if whatever the verifier learns is learnt without interacting with the prover.

An Argument system $\text{Adv}(\lambda)$ is perfectly zero-knowledge if all the adversaries within the proposed system result in zero such that $\text{Adv}(\lambda) = 0$

$$\text{R} \leftarrow \text{R}(1^\lambda), \ \text{Setup} \ (\text{R},\text{C}) \rightarrow \text{srs}; \ b \leftarrow [0,1]; \ b' \leftarrow \forall(\text{srs}); \ \text{if} \ b = b' \ \text{return} \ 1 \ \text{else} \ \text{return} \ 0$$

*4.4. Merkle Tree*

A Merkle tree is mainly a hash tree that is constructed similarly to a binary tree based on a cryptographic hash function. In the Merkle hast tree, every node is represented as a hash of the data stored such that a Node = h(data). If a node has two child nodes, it is represented as a combination of left and right child nodes such as $\text{Node}_x = \text{Node}_{x(\text{left})} + \text{Node}_{x(\text{right})}$. Therefore, the value of the parent node is satisfied with the collective hash values of its both children nodes such as $\text{Node}_{x(\text{parent})} = \text{h} \ (\text{Node}_{x(\text{left})} \ || \ (\text{Node}_{x(\text{right})})$. The Merkle hash tree brings along different benefits: (1) *Reduced Storage Cost:* The storage server is capable of verifying the authenticity and integrity of data without using the whole data content [33], (2) *Reduced Network I/O.* A small amount of data and proof is enough to check data consistency and verification [33]. (3) *Efficiecyt.* The Merkle tree is efficient enough to aggregate all the hash values of individual nodes into a single root value [33].

The proposed system works as mentioned in these steps:

Step 1: CA initializes the system.

Step 2: Prover/Vehicle requests CA for registration. Every user and vehicle needs to have a registered identity.

Step 3: Prover/Vehicle receives credentials from CA that help it to generate a zk-proof and send the proof to CA for authentication.

Step 4: CA verifies proof and aborts the authentication request if incorrect and authentication fails. Otherwise, CA generates a signature and sends it to the vehicle. User identification is stored on a Merkle tree in a blockchain.

Step 5: The blockchain sends authentication information of the vehicle to nearby RSUs.

Step 6: Vehicle i sends a message to vehicle j.

Step 7: Vehicle j signals nearby RSU to ensure if communicating vehicle i is authenticated.

Step 8: RSU responds with either a yes or no message. If authentication is True, vehicle i will start communicating with vehicle j, and the communication will be saved as a transaction by RSU monitoring the communication. The transaction will be stored on blockchain and will be distributed within the blockchain network for an update to check for a malicious data exchange.

## 5. System Construction

zk-SNARK is represented by a tuple of polynomial time algorithms $\Pi_z$ = (setup, KeyGen, Prove, Verify). Let $\complement : F^n \times F^h \rightarrow F^l$ be an arithmetic circuit and R = $\{(x, \ w)\} \subseteq F^n \times F^h$ be

the corresponding circuit satisfaction relation, where $x \in F^n$ is the statement and $w \in F^h$ is the witness.

### 5.1. System Setup

Security parameter $\lambda$ and Circuit C are the input parameters for calculating structured reference string (srs) that consists of key pairs (Pk,Vk) such that Pk is the proving key use and Vk is the verification key. Pk is used to generate zk-proof ($\pi$) and Vk is used to verify the zk-proof. The master public key $\left( K_{CA}^P \right)$ and master private key $\left( K_{CA}^s \right)$ are generated by CA. CA publishes the master public key and structured reference string 'srs' on the blockchain and keeps the master private key secret.

### 5.2. Registration

In registration phases, all CAVs and RSUs are registered with CA. The public and private keys of vehicles are generated as $\text{KeyGen}(\text{ID}_v, K_{CA}^s) \rightarrow (\text{PK}_v, \text{SK}_v)$ by CA. A similar process is done for RSU. Personal information along with the public and private keys of the vehicles is stored in the blockchain but because of zk-SNARK, the vehicles remain anonymous, and it is hard to link the user's authentication request with the identity stored in the Merkle tree. However, this aspect will help in tracing the malicious behavior of vehicles/RSUs.

### 5.3. Authentication

For the authentication of any vehicle/RSU, it is necessary to request from CA its respective leaf node '$l_{id}$' of its identity commitment, the current path 'p', and the root 'rt' of Merkle tree. This retrieved data is used to construct proof as mentioned in Algorithm 1. The proof '$\pi$' along with user identity information is sent back to CA for authentication. The construction of proof depends on the latest value of 'nu', which is a nullifier nonce that cannot be reused. After receiving the proof, CA looks for the value of 'nu' if it is updated, which means the proof is valid, but if outdated 'nu' has been used to generate proof, the authentication request is rejected. CA then uses the verify ( ) function as mentioned in Algorithm 1 to check the validity of the proof and user's authentication request. If the verification results in 'True', the status of the vehicle/RSU is broadcasted to nearby RSUs. RSU is capable of storing the result of verification or it may request blockchain every time; therefore an authentication check is received by it from some vehicle from another vehicle.

$$W = \{S(PK_i)\wedge cmt(h(S(PK_i)))\wedge(l_{id}, p)\} \tag{1}$$

$$x = (rt, nu) \tag{2}$$

$$\pi = \{srs, w, x\} \tag{3}$$

Algorithm 1 represents system system.

---

**Algorithm 1:** Setup zk-SNAKR Protocol

---

Input: Relation R, Security Parameter $\lambda$, Arithmetic Circuit C
Output: Structured Reference String (srs), Public Parameters (ppz)
$\qquad$ R←R($1^\lambda$)
$\qquad$ KeyGen (R, C) $\rightarrow$ srs
$\qquad$ srs = (Pk, Vk)
$\qquad$ return srs
end

---

Proof is generated using Algorithm 2.

---

**Algorithm 2:** Prove ( ) zk-SNAKR Protocol

---

Input: Leaf index of identity commitment ($l_{id}$), corresponding Path (p), hash function (h), Salt function (S), Public Key of Vehicle ($PK_v$), Root of Merkle tree (rt), Nonce (nu), Structured Reference String (srs)
Output: Witness w, Argument x, Proof $\pi$
   CA sends $\{l_{id}, p, h (S (PK_v))\}$ to Vehicle v
    $v \rightarrow \{ L_{id}, p, h(S(PK_v))\} = w$
   CA send rt and nu to Vehicles
    $v \rightarrow (rt, nu) = x$
     $v \rightarrow (srs, x, w) = \pi$
      Vehicle v sends $V_{id}$ and $\pi$ to CA for verification
     end
   end

---

The proof is constructed using the latest value of 'nu,' which is updated in a timely way by the blockchain network and hence cannot be forged or reused by any adversary. Vehicle v sends ($ID_v$, $\pi$) to CA; using Equation (2), CA computes x and performs verification as follows:

$$\text{Verify (srs, x, } \pi) = 1 \tag{4}$$

Verifiation process is performed in Algorithm 3.

---

**Algorithm 3:** Verify ( ) zk-SNAKR Protocol

---

Input: Root of Merkle tree (rt), Nonce (nu), Structured Reference String (srs)
Output: Argument x, Integer b
  $CA \rightarrow (rt, nu) = x$
    CA verifies the proof using srs, x and $\pi$
     if b = = 1 then
      return True
     else
      return False
end

---

Merkle Tree creation and vehicle registration is done using Algorithm 4.

---

**Algorithm 4:** Vehicle Registration and Merkle tree Creation

---

Input: Root of Merkle tree (rt), Leaf index of identity commitment ($l_{id}$), corresponding Path Map (p), Identity of Vehicle ($ID_v$), Hash function (h), Salt function (S), commitment (cmt), Vehicle V = $\{a_1, a_2, a_3, \ldots, a_l\}$
Output: Public Key of vehicle ($PK_v$) and Private Key of Vehicle ($SK_v$), Merkle tree node ($MT_{node}$), Root value (rt)
  Vehicle v sends its $V_{id}$ to CA
  $CA \rightarrow (V_{id}, K_{CA}^s) = (PK_v, SK_v)$
    $MT_{node}$ consists of $cmt(h(S(PK_v), V_{id})$
     $MT_l = MT(a_1, a_2, a_3, \ldots, a_{\lceil n/2 \rceil})$
     $MT_r = MT(a_{\lceil n/2 \rceil + 1}, a_{\lceil n/2 \rceil + 2}, \ldots, a_l)$
     $h(rt) = h(MT_l \mid\mid MT_r)$
     Update rt
    Return (rt, p and $l_{id}$) to Vehicle v
    End

---

### 5.4. Communication

Once a vehicle is authenticated on a blockchain network, it can start communicating with other vehicles. $V_i$ computes its signature first and sends the signature as token of the initiating communication to vehicle j as follows:

$$\sigma_i = (\pi \mid \mid SK_i \mid \mid T)^{PKi} \tag{5}$$

'T' denotes the timestamp in order to justify if the message is new or old message and is being circulated by the vehicle. $V_j$, once receiving the signature, instead of opening it, sends an authentication check message to nearby RSU. RSU would check the authentication status of $V_i$ either with data stored with itself, or by transferring the check request to the blockchain. Once RSU receives the check results, it sends a 'True/False' message to the requesting vehicle $V_j$. If the status is 'True', $V_j$ starts communication with $V_i$ and rejects messages from it altogether. The acceptance of the message from $V_i$ is responded to by $V_j$ using a signature of $\sigma_j = (\pi \mid \mid SK_j \mid \mid T)^{PKj}$.

### 6. Performance Analysis

In this section, security on privacy preservation, resistance against a single point of failure, and DDoS is analyzed in addition to the simulation of the proposed system. We evaluate the performance of the proposed system and compare it with the related architectures proposed in the literature in terms of the average packet delay, average packet loss, average packet delivery in the presence of DDoS attack, and also with/without an Eclipse attack on the blockchain. The configuration of the parameter settings is presented in Table 2. A local Ethereum blockchain network is implemented based on PoA on a Lenovo Laptop with the following parameters: Intel Celeron CPU N2840 @ 2.16 GHz, 2 GB RAM). Parity nodes being implemented to connect to RSUs in NS2. zk-SNARK are run where the proof is generated by the vehicle/RSU (Prover) and is verified by CA (verifier). A Zokrates toolbox is used to implement Equations (1)–(3) and sha256 for hashing using MIMC. The Pproof generation time is almost 5 s and proof verification time is 3 milliseconds.

**Table 2.** Configuration Parameters.

| Parameters | Settings |
| --- | --- |
| Simulation Time | 30 min |
| No. of nodes | 500 |
| Max speed of nodes | 20 m/s |
| Node Mobility | Random |
| Area size | 1000 m * 2500 m |
| Traffic | Constant Bit Rate (CBR) |
| Distance between nodes (average) | 2.5 m |

### 6.1. Privacy Preserving

Anonymity and Unlinkability: Anonymity and unlinkability have been successfully achieved in the proposed system through anonymous credentials using zk-SNARK. The real identities of vehicles are masked so that not even RSU can relate a vehicle to a particular identification that satisfies the first design goal. Secondly, with the use of the updated nullifier 'nu', a vehicle's driving trajectory/history is hard to be traced since CA rejects all requests from those who try to forge the proof generated by another vehicle using its 'nu' in order to trace the driving aptitude and location of a specific vehicle. If a registered vehicle/RSU behaves maliciously, it is easy for the CA to locate its identity and ban it from the proposed system. The interesting thing about this proposed system is that proof is generated only once by every vehicle/RSU during its registration and authentication phase, which reduces the computational cost of computing the zk-SNARK proof and makes the system quite lightweight and quick in terms of proof generation/verification.

### 6.2. Security against SPoF and DDoS/Eclipse Attacks

With the blockchain, the problem of single point of failure (SPoF) is omitted altogether. Blockchain is a decentralized architecture composed of different nodes such as RSUs, so it becomes very easy for vehicles within the network to access the network if even a single RSU is working honestly within the network. Single point of failure is effectively omitted through the proposed system by using blockchain and hence satisfies the second design goal. The second design goal is the major intention of this study. RSUs and vehicles registered within the blockchain network remove the illusion of DDoS and Eclipse attacks from adversaries in the shape of a malicious RSU or vehicle. Figure 4 shows the overview of the attack scenario within the proposed system. The system is secure against DDoS and Eclipse attacks as the unregistered nodes have been treated as malicious nodes that can try to communicate with registered nodes using their signature. Since the signature is generated as $\sigma = (\pi \mid\mid SK \mid\mid T)^{PK}$ and only registered vehicles can compute the secret proof '$\pi$', this means an incomplete signature is sent from the malicious node to the registered node. The receiving node can easily identify the incomplete signature and report the malicious node to the nearby RSU who can block its activities from the network. The same may happen with RSU if any malicious node tries to send malicious data to the RSU, but because of an incomplete signature, its messages are not going to be delivered. Tracing malicious nodes can be effectively achieved by identifying unregistered and unauthenticated nodes trying to communicate with other nodes within the network, thus satisfying the third and fourth design goals.

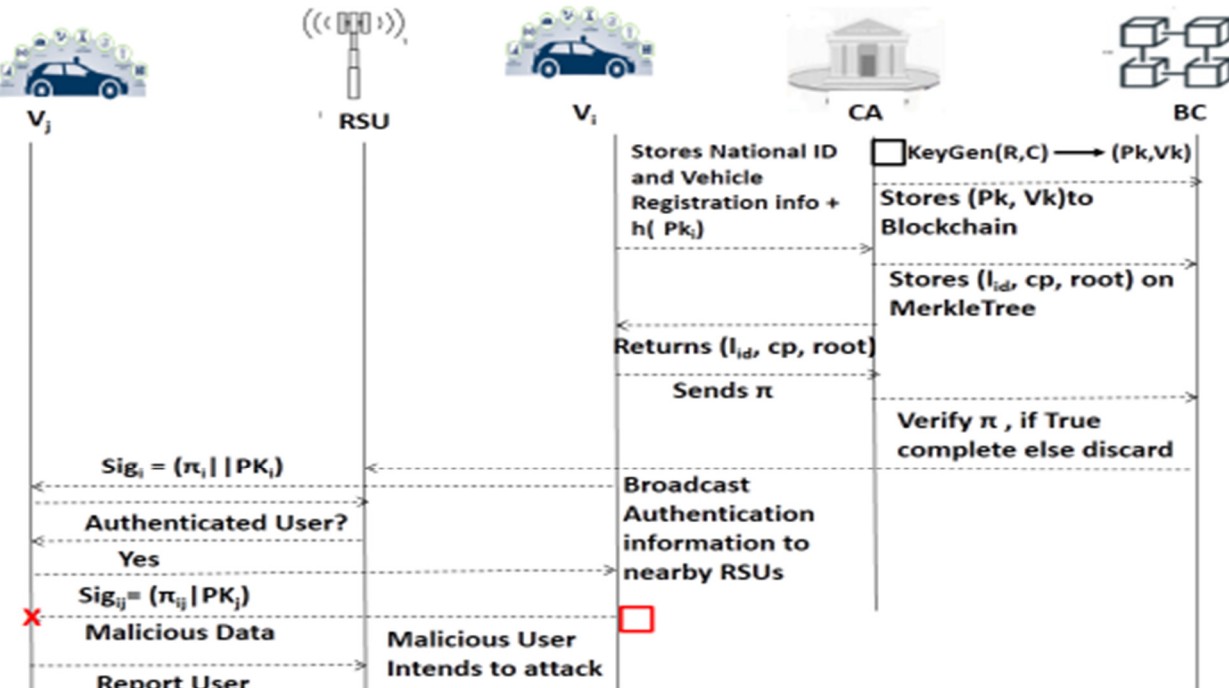

**Figure 4.** Overview of the Attack Scenario within Proposed System.

#### 6.2.1. Average Packet Delay Ratio

Average packet delay is the time taken by a packet to reach to its destination from the source. Figure 5 shows the results of an average packet delay ratio in the presence of a DDoS attack. Figure 5a shows a steady average packet delay in the presence of a DDoS attack with respect to the number of nodes as opposed to Bilinear Pairings [46], Secure Trusted-based Blockchain [25], and Hybrid Approach [47]. Figure 5b shows the average packet delay ratio in the presence of a DDoS attack with respect to time. It can be observed from both Figure 5a,b that the proposed system brings steady or minor changes in the average packet delay ratio in contrast with others.

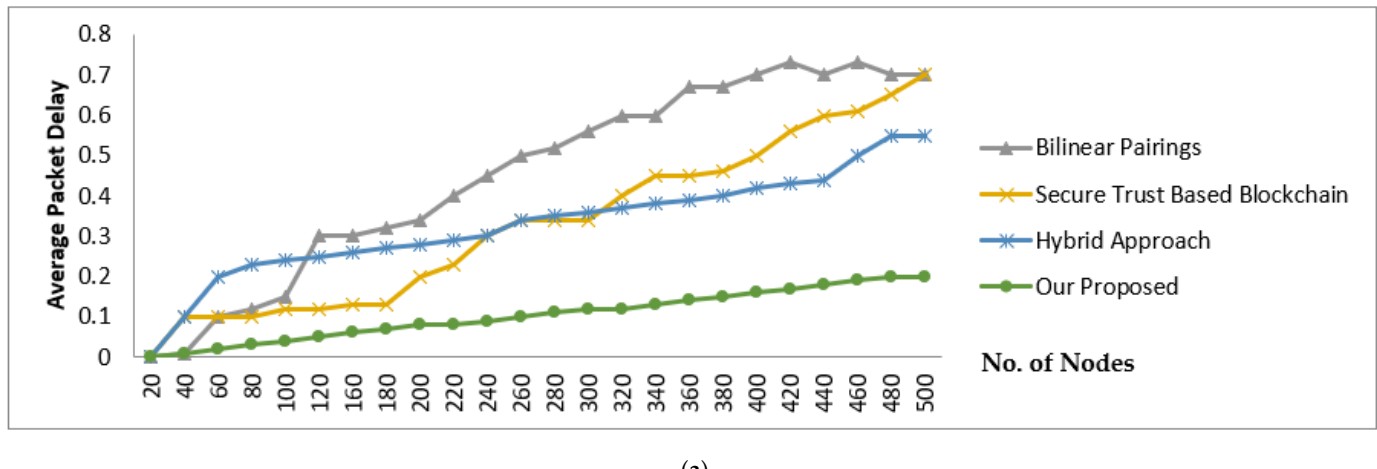

(**a**)

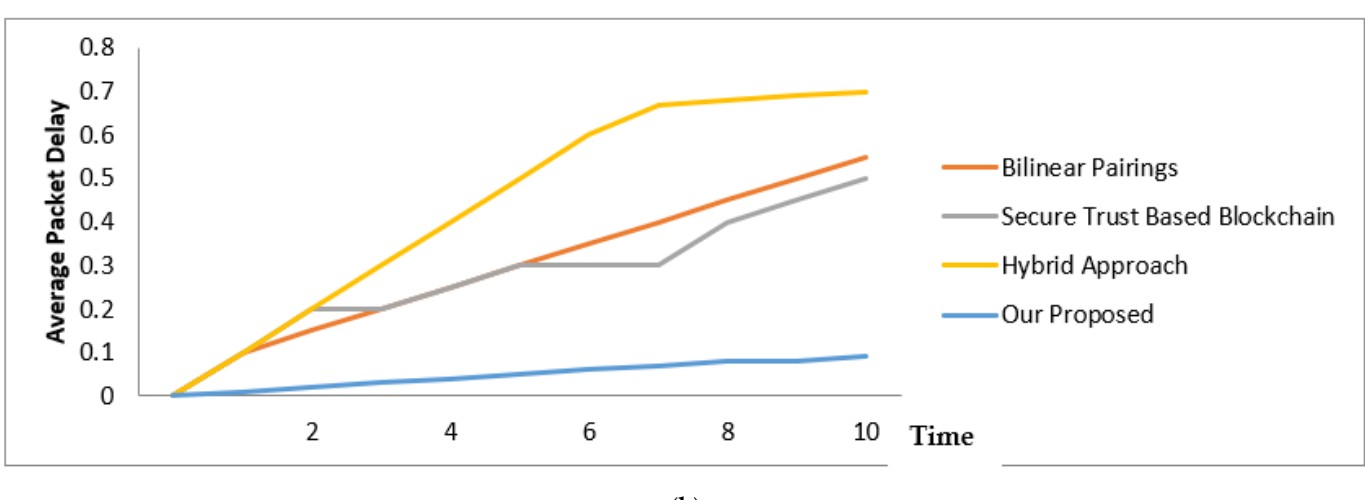

(**b**)

**Figure 5.** (**a**): Average Packet Delay with DDoS Attack w.r.t No. of Nodes; (**b**): Average Packet Delay with DDoS Attack w.r.t Time.

### 6.2.2. Verage Packet Loss Ratio

Packet loss is the proportion of packets lost/dropped against total packets sent in the network. Figure 6a shows that the proposed system suffers from less average packet lost in the presence of a DDoS attack with respect to number of nodes and Figure 6b with respect to time as opposed to Bilinear Pairings [46], Secure Trusted Blockchain [25], and Hybrid Approach [47]. With the increased number of nodes, the efficiency of the one-time zero-knowledge proof generation scheme in the proposed system makes the packet loss ratio go steady.

### 6.2.3. Average Packet Delivery Ratio

The number of packets successfully delivered to the destination to the total number of packets sent is known as the packet delivery ratio. Figure 7a shows that the packet delivery ratio is higher in the proposed system in the presence of a DDoS attack with a slight decrease as the number of nodes starts increasing from 50 to 90, but becomes steady when the number of nodes continues, increasing further in contrast to other proposed architectures. Figure 7b shows the Average Packet Delivery Ratio in the presence of DDoS with respect to time.

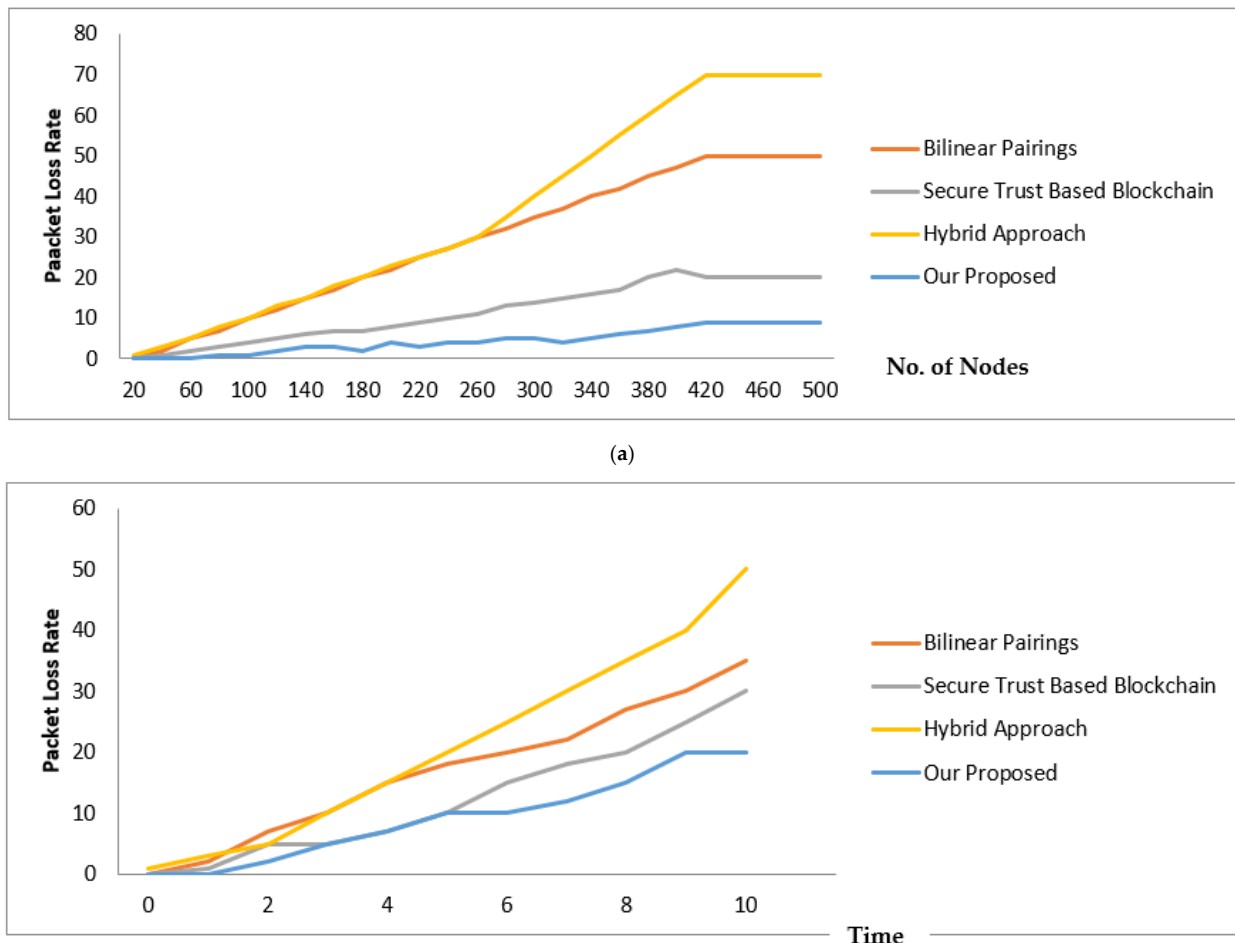

(**a**)

(**b**)

**Figure 6.** (**a**): Average Packet Loss Rate with DDoS Attack w.r.t No. of Nodes; (**b**): Average Packet Loss Rate with DDoS Attack w.r.t Time.

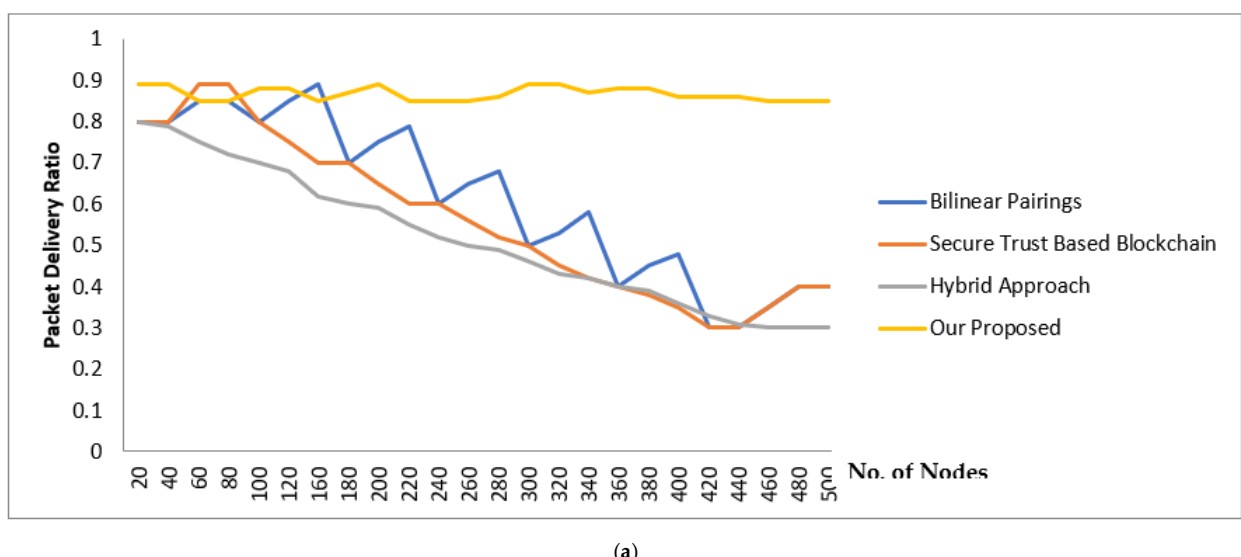

(**a**)

**Figure 7.** *Cont.*

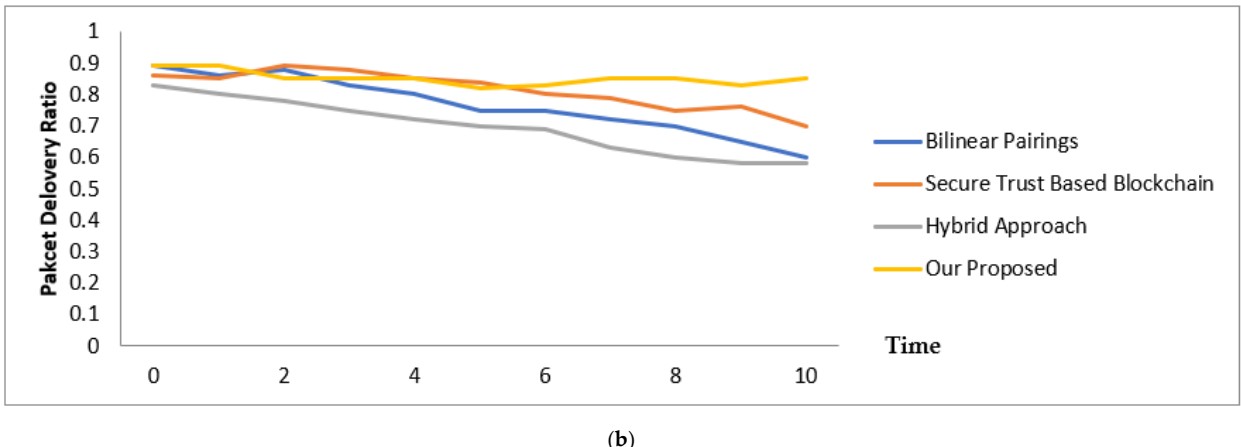

(**b**)

**Figure 7.** (**a**): Average Packet Delivery Ratio with DDoS Attack w.r.t No. of Nodes; (**b**): Average Packet Delivery Ratio with DDoS Attack w.r.t Time.

### 6.2.4. Average Delay with Eclipse Attacks

The proposed system has been evaluated for its performance against an Eclipse attack on the blockchain structure. Figure 8a shows an average packet delivery delay in the absence of an Eclipse attack and Figure 8b shows an average packet delivery delay in the presence of an Eclipse attack. From Figure 8b, it is observed that there is a slight delay in the packet delivery when an Eclipse attack is launched against blockchain but the difference is minor.

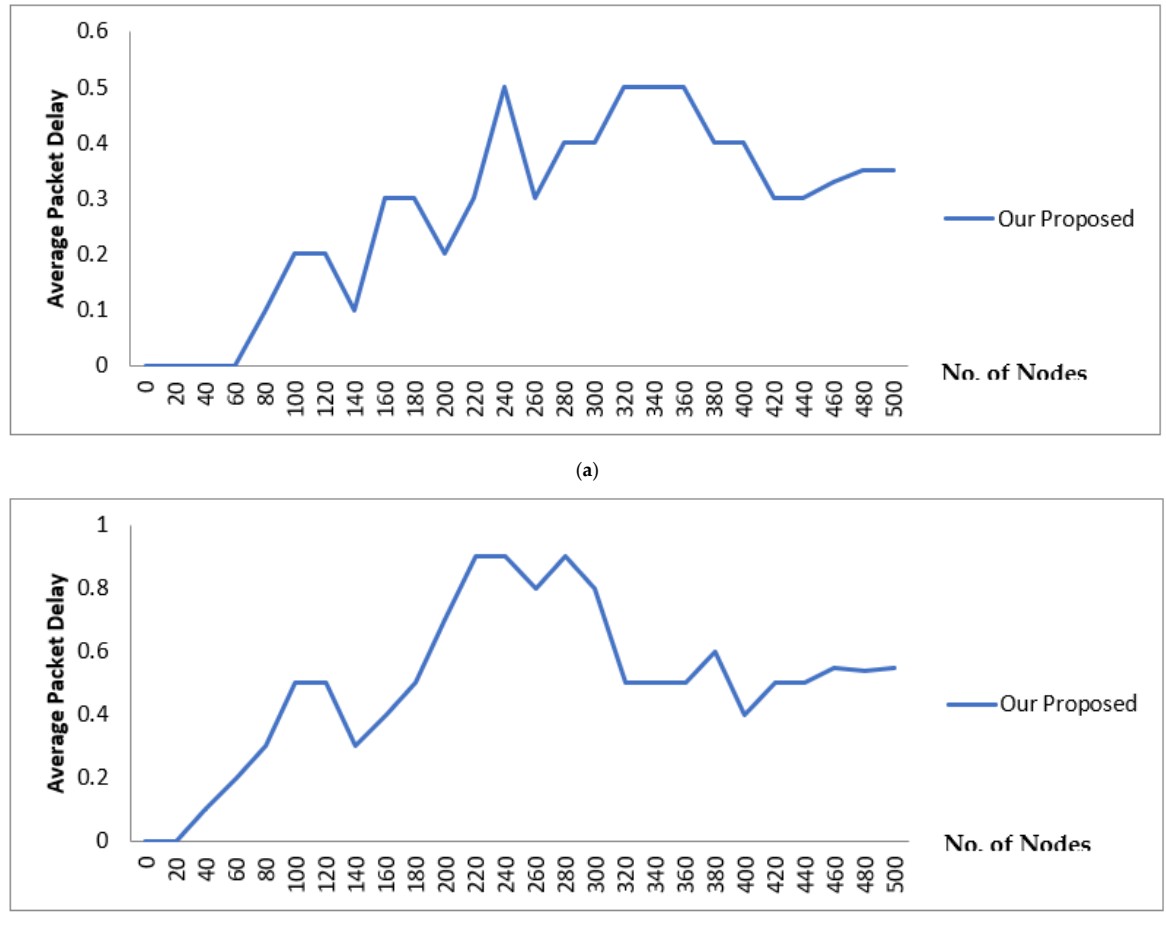

(**a**)

(**b**)

**Figure 8.** (**a**): Average Delay without Eclipse Attack; (**b**): Average Delay with Eclipse Attack.

## 7. Discussion

Figure 5a,b shows that the proposed system incurs a steady packet delay with an increasing number of nodes whereas other models show sudden hikes in packet delay with more nodes being added. Particularly increasing nodes beyond 200 makes other models show greater delays. This is due to the fact that light-weight hash and Salt functions have been used to generate the proof only once for every node. The simulation results show that packets were quickly processed in a short time as compared to other models.

From Figure 6a,b it is evident that the packet loss of the proposed system is much lower with the increasing number of nodes, although slight jumps have been observed between 300–400 but they turn to steady with further increases. On the other hand, the other benchmarks used do not perform well in their performance with an increasing number of nodes. This is because the zk-SNARK is generated once, reducing the time of computation and making the algorithm stronger against the attack. The other models are not strong nor capable enough to protect the network from a DDoS attack with the increasing number of nodes, although the result of all the compared models are almost same with a minor number of nodes.

With the increase in the number of nodes, the decrease in the packet delivery is observed in the presence of a DDoS attack as shown in Figure 7a,b. This trend is found in general in all cases. However, we observed that rest of the models (used as benchmarks for performance comparison) had a limited number of nodes between 100–200. The simulation results, however, show an increasing number of nodes beyond 200, and the performance of other models show a steady decrease in average packet delivery. This communication channel becomes congested with both the malicious and valid nodes trying to transmit data across the network. A higher packet drop is observed in all the cases including ours once the number of nodes increases but this impact turns steady in our proposed model and the packets drop becomes lesser. This is because our zk-SNARK algorithm takes less time in computation and is less intense in calculations, which helped in reducing the packet delay and as a result results in higher packets delivery as compared to other models. The compared models showed a drastic slump in packet delivery when the number of nodes started increasing from 80, with their performance ultimately deteriorating, the justification being the algorithms used in the other models were not capable of protecting the communication channel with an increasing number of nodes in the presence of DDoS and, hence, resulted in a higher loss of packets.

Most of the security models presented in the literature are based on assumptions, i.e., the majority of the nodes within the blockchain network, particularly RSU, are honest [33] and not compromised, but our model does not work on any sort of honest assumption. This helps in devising a broader threat model. Since blockchain is open for all the members on the blockchain network to view the transactions, it can reveal the privacy of users/vehicles and also help in tracking their location. The threat model for the proposed system is as follows:

Threat (1) Vehicles Showing Malicious Behavior: A vehicle may try to avoid authentication within the blockchain network and may attempt to send messages to other vehicles. Messages from a vehicle are sent along with the normal identification of the vehicle such as (VID), which is the registration number/vehicle plate number and digital signature of vehicle once they are rejected. The transmitted message should be concealed within the digital signature and must include the 'PK' of the vehicle and the proof '$\pi$' in order to be verified by the blockchain. Therefore, any message containing simply 'VID' concealed within the digital signature will be rejected straightaway. Malicious vehicles can also send messages containing 'PK' but missing the proof '$\pi$' concealed within the digital signature of the vehicle. This means a vehicle has not been registered within the network and is not trying to communicate with other vehicles; therefore, messages with missing '$\pi$' are also rejected straightaway.

Threat (2) RSUs Showing Malicious Behavior: RSU is also not assumed to be honest in our system, so any RSU can try to behave abnormally. Cyber attackers may try to

compromise RSU by remotely hijacking it. It can verify the authentication of any malicious vehicle without verifying it from the smart contract. Secondly, it can itself send malicious messages intended by the attackers to vehicles to distract them. Cyber attackers can launch DDoS and Eclipse attacks through RSUs easily. This threat is also mitigated by the proposed system as every RSU is also required to be registered with 'CA' before being part of the network. This means an RSU must have to generate the proof 'π,' which should be verified by the blockchain and the result of verification is then distributed within the network in order to let other RSUs and vehicles know the status of specific RSU [48,49].

Threat (3) Disclosure of Vehicle Identity: Because of the open nature of the blockchain, the identity information of vehicles may be inferred from the blockchain. Owing to linkability This can help adversaries download the full trajectory history of specified vehicle from a blockchain to perform statistical analysis to expose the traveling history and traveling habits of a vehicle. The linkability threat has been mitigated by the proposed model by using the zk-SNARK protocol, which hides the real identities of the communicators.

## 8. Conclusions and Future Work

Security and privacy has always been an issue when the distribution of computation has to take place in real scenarios. CAV is the best example of such a scenario where vehicles are connected with one another and everything along the road, sharing data/information consistently. The autonomous nature of connected vehicles makes them more independent when making decisions on their own after perceiving the environment. Any malicious entity within the surroundings can damage the traveling habit of users travelling in CAVs. In this paper, a security and privacy scheme for CAVs has been proposed on blockchain. The proposed system utilizes the zk-SNARK protocol to preserve the privacy of CAVs in addition to ensuring unaltered communication through blockchain. It successfully achieves secure communication among multiple CAVs in addition to concealing the real identities of the communicators. It has managed to resist SPoF, Eclipse, and DDoS attacks and is capable of tackling/identifying malicious CAVs/RSUs. Blockchain has been an effective technology to achieve security within the IoV, which can be deployed in different scenarios depending on the requirements. The same has been achieved in this paper by cashing in on the features of the unlinkability and immutability of blockchain along with the confidentiality and anonymity of zk-SNARK for preserving the privacy of communicators. The extension to this study will be conducted by removing the Salting function in order to assess the privacy preserving feature of zk-SNARK alone and compare the performance of the new model with this proposed model.

**Author Contributions:** R.K. and A.M. conceived of the presented idea. A.M. developed the theory and performed the computations. A.M. and Z.I. verified the analytical methods. A.M., C.M. and G.E. encouraged R.K. to investigate the blockchain and its attack vectors and supervised the findings of this work. All authors discussed the results and contributed to the final manuscript. R.K. and Z.I. carried out the experiment under the supervision of C.M., A.M. and R.K., who wrote the manuscript with support from A.M., C.M. and G.E., A.M. and C.M. conceived the original idea. G.E. developed the theoretical formalism, performed the analytic calculations, and performed the numerical simulations. All authors (R.K., A.M., Z.I., C.M. and G.E.) contributed to the final version of the manuscript. A.M. supervised the project. R.K., A.M., Z.I., C.M. and G.E. conceived and planned the experiments. R.K., A.M., Z.I., C.M. and G.E. conducted the experiments. R.K., A.M., Z.I., C.M. and G.E. planned and carried out the simulations. R.K., A.M., Z.I., C.M. and G.E. contributed to the interpretation of the results. R.K. took the lead in writing the manuscript. All authors provided critical feedback and helped shape the research, analysis, and manuscript. R.K. and A.M. designed the model and the computational framework and analysed the data. R.K. and A.M. wrote the manuscript with input from all authors. C.M. and G.E. conceived the study and were in charge of overall direction and planning. All authors have read and agreed to the published version of the manuscript.

**Funding:** This research received no external funding.

**Institutional Review Board Statement:** Not applicable.

**Informed Consent Statement:** Not applicable.

**Data Availability Statement:** Not applicable.

**Conflicts of Interest:** The authors declare no conflict of interest.

## Appendix A

**Table A1.** Summary of the Literature Study.

| Ref | Types of Attacks Being Handled | Security Objective | Advantages | Limitations | Simulator/ Computation/ Analysis Tools |
|---|---|---|---|---|---|
| [10] | • DoS attack<br>• Collusion Attack<br>• Eavesdropping attack<br>• Replay attack<br>• Impersonation attack<br>• GPS Spoofing | • Data integrity<br>• Non-repudiation<br>• Confidentiality | • Reduces latency and complexity of certificate based PKI | • Assumptions that vehicles and RSUs are equipped with Hardware Security Module (HSM) | Proverif |
| [11] | • Impersonation attack<br>• DDoS attack | • Secure data transmission<br>• Confidentiality<br>• Integrity<br>• Non-repudiation | • Message transmission through trusted vehicles<br>• Vehicle revocation on the basis of reputation value | • Focuses on V2V communication only | Not Available |
| [12] | • Message injection attack with Spoofed ID<br>• Fabrication attack<br>• Suspension attack<br>• Masquerade attack | • Security<br>• Safety | • Fingerprints the malicious ECUs | • Assumptions are made on clock behavior | CAN bus prototype and real vehicle |
| [13] | • Message injection attacks<br>• DoS attack | • Safety of drivers and passengers | • The proposed system is simple to use, consuming less computing power | • The malicious messages are assumed to be sent in shorter time intervals though it may happen in longer time intervals | K-car is used as the testing vehicle, KVASER CAN interface is used to connect to the CAN bus |
| [14] | • Forgery<br>• Impersonation attack<br>• Man in the middle attack<br>• Replay attack | • Data security<br>• Privacy preserving | • Identity verification is quite intensive task requiring verification of everything within the blockchain network and guarantees authentic data source | • Traditional encryption techniques are used, asymmetric and symmetric encryption are time consuming | Not Available |
| [15] | • Impersonation attack | Privacy preserving | • The system does not require any registration and authentication<br>• System works as an app | • Policies and rules to monitor credit-sharing are not concrete<br>• K-anonymity encryption technique works on probability and, hence, is not completely secure | Python provided by Eric Alcaide |

**Table A1.** *Cont.*

| Ref | Types of Attacks Being Handled | Security Objective | Advantages | Limitations | Simulator/ Computation/ Analysis Tools |
|---|---|---|---|---|---|
| [16] | • False information attack<br>• Sybil attack | Intrusion detection | • Statistical techniques are used to detect anomalies and malicious vehicles using traffic model | • Depends on consensus mechanism where vehicles share their flow and speed values with each other in order to build the traffic scenario<br>• The coordination of rogue nodes (malicious vehicles) is important and if their communication pattern varies at large, the system may not be effective to detect the attack | OMNET++, SUMO and VACaMobil |
| [17] | • Prevent/lower accident cases | Secure data sharing | • Secure and real time accident information sharing | • Vehicle registration and authentication procedure are not provided<br>• Works on consensus, which may involve many malicious vehicles to propagate wrong information | Not Available |
| [18] | • Eavesdropping attack | • Preventing data loss<br>• Secure communication | • Probabilistic nature of the Bloom filter is handled by limiting number of hash functions<br>• Bloom filter helps in reducing the data transmission | • The validation time of the system increases with the increase in number of vehicles<br>• The processing time of hash functions also increases with an increase in number of users | Matlab and Python 3.7 (google colab) |
| [19] | • Fabricated data<br>• Data collusion | Data privacy | • Federated Learning involving Machine Learning Model | • The model is mainly used to retrieve data similar to a search engine but does not explain the identity privacy mechanism | Testing on two real world data sets:Reuters dataset and 20 Newsgroups dataset |
| [20] | • Linking Attacks<br>• Malware<br>• Falsified Attacks<br>• DDoS | Secure Communication | • Optimized for large scale low resource networks<br>• Distributed trust algorithm to reduce the processing time connected with every validating blocks | • A vehicle may have changeable private keys during its lifetime that can increase burden on relay nodes<br>• Asymmetric encryption is comparatively slower | Not Available |
| [21] | • Impersonation Attacks<br>• Falsified Attacks<br>• Attacks on Ports | Secure Communication | vehicle may not need to share entire dataset | • Complete privacy is not ensured | Not Available |

**Table A1.** *Cont.*

| Ref | Types of Attacks Being Handled | Security Objective | Advantages | Limitations | Simulator/ Computation/ Analysis Tools |
|---|---|---|---|---|---|
| [22] | • Random packets injection<br>• Impersonation attacks<br>• DDoS attacks | • Secure communication<br>• Preserving privacy | • Decentralised KPI<br>• Self-Pseudonyms generation by vehicles | • KPI generation/verification is time consuming<br>• Conditional privacy | Not Available |
| [23] | • Data modification | Privacy preserving | • Reliable data source using blockchain | • Assumption TA is trustworthy | MacBook Pro, 2.3 GHz, Corei5 processor and 8GB 2133 MHz LPDDR3 Memory |
| [24] | • Impersonation attacks<br>• Sybil attacks | Identity and Location Privacy | • No reliance on Centralized Authority (CA)<br>• Multiple sub-identities for vehicles | • K-Anonymity Unit is NP hard<br>• Processing time is enhanced if more vehicles are added to VANET | OPNET and Ethereum |
| [25] | • Sybil attacks<br>• DDoS attacks | • Privacy preserving<br>• Security of users | • Emergency vehicles are given priority<br>• Light-weight RSA-1024 digital signature algorithm is utilized, which takes less time for authentication as compared to ECDSA | The authentication process is still time consuming because of the classification of vehicles | Ganache |
| [26] | • Sybil attacks<br>• DoS<br>• Impersonation attacks<br>• Data modification attacks | • Privacy protection<br>• Trust management<br>• Security of users | • Freshness of messages is ensured using time stamps and hashing technique<br>• Message rating technique is used to manage trust | The authentication process is quite lengthy and requires one vehicle to consult many blockchains for verification of another vehicle, which takes a lot of time | Veins |
| [27] | • Eavesdropping attacks<br>• Replay attacks<br>• Sybil attacks<br>• Location information protection | • Privacy protection<br>• Location protection | • Secure user registration<br>• Efficient key management<br>• Two-way authentication | • Based on consensus algorithm Proof of Work (PoW) that consumes a lot of computing power<br>• Key exchange procedure also results in storage burden | Intel Core i5-9400f CPU with2.90 GHz, 16 GB RAM, Win10 processor |
| [28] | • Spoofing attacks<br>• Eavesdropping attacks | • Data integrity<br>• Pprivacy preserving | Use of Gateway to switch from one Blockchain to another | • Vehicle registration process is not mentioned<br>• Describes only communication of vehicles with Gateways and V2V communication are not entertained | Hyperledger Fabric v1.2 and Hyperledger Ursa, JSFiddle for Reverse Geocoding and Hyperledger Caliper for Benchmark tests |
| [29] | Double spending attacks | • Privacy<br>• Fairness<br>• Unlinkability | Ensures privacy preserving through zk-SNAKR | Zk-SNARK has been used excessively, which means proof generation and verification consumes a lot of computation resources, making the system ineffective and slow | Ethereum, open source library libsnark |

**Table A1.** *Cont.*

| Ref | Types of Attacks Being Handled | Security Objective | Advantages | Limitations | Simulator/ Computation/ Analysis Tools |
|---|---|---|---|---|---|
| [30] | DoS | • Authenticity<br>• Integrity<br>• Availability<br>• Reliability<br>• Security<br>• Privacy | • Secure transmission of firmware updates<br>• Proof of distribution is achieved through zk-SNARK, which preserves the privacy of vehicles | Use of ABE to justify the firmware update access policy, but ABE is a time-consuming traditional encryption technique, which can make the firmware update process quite slow | Intel Core i7-4765T 2.00 GHz and 8 GB RAM, Python charm cryptographic library for zk-SNARK |
| [31] | • Offline password guessing attacks<br>• Replay attacks<br>• Impersonation attacks<br>• DDoS | • Secure authentication<br>• Anonymity (privacy) | • Accuracy and trustworthiness of messages<br>• Protects vehicles privacy | • Vehicle's OBU is assumed to be temper-proof<br>• ABE is a time-consuming and traditional encryption technique | Laptop running Ubuntu 18.04 OS, Intel Core i7-6700 CPU with 3.40 GHz, 2 GB RAM, cpabe-toolkit for ABE scheme |
| [32] | • Impersonation attack<br>• Data Falsification<br>• DDoS<br>• Man-in-the-middle | • Safety<br>• Transparency | Safe cab ride through voting | • Vehicles' numbers and vehicle/user ratings are stored in ordinary tables as well as on the blockchain but data can be tracked/hacked and altered from ordinary tables<br>• User/vehicle communication is done using real identities, which can be forged easily | NS2 |
| [33] | DDoS | • Authentication<br>• Privacy<br>• Reliability<br>• Availability<br>• Scalability | Multi-agent system to automatically guide traffic | Any vehicle sending a request to IM is considered as legitimate vehicle through the authentication process, but in case the credentials of vehicles are hacked, there is no mechanism to tackle the situation. The privacy of vehicles' credentials is not mentioned | Hyperledger Fabric |
| [34] | • Free-riding attacks<br>• Double-claim attacks<br>• DDoS attacks<br>• SPoF<br>• Repudiation attacks<br>• Unforgeability | • secure communication<br>• Preserving privacy | Ad dissimination is honestly achieved | Based on assumptions | Ethereum Blockchain environmet is setup in Lenovo desktop (Intel Core i5-3470 3.20 GHz Quad Core CPU, 8 GB RAM), VANETSim |
| [35] | • Data theft<br>• Identity forgery<br>• Replay attacks<br>• Masquerade attacks | • User Privacy<br>• Vehicle Privacy | One-way authentication model | Side channel attacks are possible | Not Available |

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
