# Peer review of "Security and Privacy in Connected Vehicle Cyber Physical System Using Zero Knowledge Succinct Non Interactive Argument of Knowledge over Blockchain"

_applsci, doi:10.3390/app13031959_

Round 1

Reviewer 1 Report

Authors proposed a Zero Knowledge based architecture for Vehicle. However actual implementation of ZK-Snark with this is not that much straight forward and easy. 

Most of the graphs are not actual and I don't think they are correct. However please provide experimental setup and code on Github so that we can verify the graphs.

Article is poorly written and many notations are not clear. Provide a table for notations only. 

Many concepts presented in the paper are wrong.

Formatting of the paper is very poor. 

Overall I think this paper is just concept and I don't think authors have actually done some experimental work.

Author Response

Comment:
The authors proposed a Zero Knowledge based architecture for the vehicle. However the actual implementation of ZK-Snark with this is not that much straight forward and easy. 

Response:
The Implementation was not easy and that took too much of our time almost a year to make the proposed architecture into working

Comment:
The article is poorly written and many the notations are not clear. Provide a table for notations only. 
Response:
The article's language and grammatical mistakes have been addressed. The notations have been made clear and have been provided in the table

Comment:

Most of the graphs are not actual and I don't think they are correct. However, please provide experimental setup and code on Github so that we can verify the graphs.

Response:

We requested our institute for that, we are waiting for their reply, but not sure we get their positive reply. On the other hand, we are ready to justify that our graphs are correct because we have checked it double. 

Comment:
Many concepts presented in the paper are wrong.
Response:
The article has been revisited in order to correct those and corrected some of them. It would be helpful if the reviewer could please specifically mention those, if not corrected will be corrected accordingly. 

Comment:
Formatting of the paper is very poor. 
Response:
Yes, the formatting mistakes have been rectified

Reviewer 2 Report

It is a substantial issue, and the paper addressed the problems and previous literature well. The analytic model is clear, and the solution is appropriate to realize. Minor revision is recommended for the paper's better quality, such as typos and citation updates. 

Author Response

Comments:
It is a substantial issue, and the paper addressed the problems and previous literature well. The analytic
model is clear, and the solution is appropriate to realize. Minor revision is recommended for the paper's
better quality, such as typos and citation updates. 
Response:
All the grammatical revisions have been addressed and citations are updated accordingly. 

Reviewer 3 Report

Refer to the attached review report.

Author Response

Comments:
There are some grammar mistakes and text that need to be rewritten (e.g., "is am encryption protocol",
"to ours is [35] and [42]")
Response:
The grammatical mistakes have been corrected. 

Reviewer 4 Report

There are some grammar mistakes and text that need to be rewritten (e.g., "is am encryption protocol", "to ours is [35] and [42]")

Some parts of the literature review are confusing, as the authors present sequences of several sentences in long paragraphs where each sentence references a different work. This section could be rewritten to make it clearer. 

In Section 2, there is a reference for Table II, but there is no table II in the work. It would be interesting to have a description of the evaluated attacks.

Section 2 states Table A1 presents the summary of the literature study. But there are missing works in such Table. For instance, works [35] and [42], which are the ones that authors claim to be the most similar to theirs, aren't referenced in Table A1.

The quality of the figures must be improved. In Figure 1, there is some unreadable text. The resolution of Figures 2 and 3 should be improved. Graphs should be of the same size or at least the graphs that share the same number (i.e., 5a and 5b, 6a and 6b, 7a and 7b) should be of the same size to improve presentation. The series of values in Figures 8a and 8b should be on a single graph.

In Section 3, the authors state that their model has three major elements, but they list four elements.

Most of the text in Section 4.1 would be better fitted in a background section. Section 4.2 has only one paragraph. It's too small to be a section. 

All the functions and parameters used in Section 4.3 to show the system's characteristics and in Section 5.1 should be clearly defined.

Authors should review the Titles of Axises in Graphs (e.g., in Figure 7.b "Pakcet Delovery").

The simulation environment and parameters are not described.

Section Discussion does not discuss the experimental results. It does not show how the proposals outperform the ones of other works.

All the works in References should be referred to in the text.

Author Response

Some parts of the literature review are confusing, as the authors present sequences of several sentences
in long paragraphs where each sentence references a different work. This section could be rewritten to
make it clearer. 
In Section 2, there is a reference for Table II, but there is no table II in the work. It would be interesting to
have a description of the evaluated attacks.
Section 2 states Table A1 presents the summary of the literature study. But there are missing works in
such Table. For instance, works [35] and [42], which are the ones that authors claim to be the most
similar to theirs, aren't referenced in Table A1.
Response:
Literature Review has been revisited to make it more clear and understandable. Table II had been renamed as Table A1 but, mistakenly, it was not removed from text. Reference [35] and [42] has been added to the table A1 and the sequence has been updated as [34] and [35] respectively

Comment:
The quality of the figures must be improved. In Figure 1, there is some unreadable text. The resolution of
Figures 2 and 3 should be improved. Graphs should be of the same size or at least the graphs that share
the same number (i.e., 5a and 5b, 6a and 6b, 7a and 7b) should be of the same size to improve
presentation. The series of values in Figures 8a and 8b should be on a single graph.
Response:

Quality of images has been improved and the resolution of the graphs has also been improved to make them all of the same sizes. Figure 8a and 8b couldn’t be added to same graph due to bad resolution result so they are placed separately.

Comment:
In Section 3, the authors state that their model has three major elements, but they list four elements.
Response:
It is updated as FOUR, was mistakenly written Three

Comment:
Most of the text in Section 4.1 would be better fitted in a background section. Section 4.2 has only one paragraph. It's too small to be a section. 
Response:
These sections are just an introduction about basic components used in the system, so they are not given detailed discussion

Comment:
All the functions and parameters used in Section 4.3 to show the system's characteristics and in Section
5.1 should be clearly defined.
Response:
Parameters mentioned in 4.3 are generally in terms of zk-SNARK however, they have been removed from the text in order to avoid confusion

Comment:
The simulation environment and parameters are not described.
Response:
Simulation Environment and parameter list has been added in Table 2 in the text

Comment:
Section Discussion does not discuss the experimental results. It does not show how the proposals outperform the ones of other works.
Response:
The experimental results have been discussed in the performance evaluation section while Discussion section explains how the threat model has been treated by the proposed system

Comment:
All the works in References should be referred to in the text.
Response:
References have been adjusted and properly referred in the text as suggested.

Round 2

Reviewer 1 Report

Authors updated the paper and replied our queries. Some of the important and recent papers are missing such as:  Privacy preserving authentication system based on non-interactive zero knowledge proof suitable for Internet of Things,  Privacy-preserving ledger for blockchain and Internet of Things-enabled cyber-physical systems etc.

Author Response

The change is reflected on page numbers 9,10, 11 and also Reflected in the References section

Reviewer 4 Report

The presentation is still poor. Some images remain with unreadable text. 

Author Response

The diagrams are fixed on pages 8,10 and 16 to make them better readable